# Cardiac Output and Cerebral Oxygenation in Term Neonates during Neonatal Transition

**DOI:** 10.3390/children8060439

**Published:** 2021-05-24

**Authors:** Nariae Baik-Schneditz, Bernhard Schwaberger, Lukas Mileder, Nina Höller, Alexander Avian, Berndt Urlesberger, Gerhard Pichler

**Affiliations:** 1Division of Neonatology, Department of Paediatrics and Adolescent Medicine, Medical University of Graz, 8036 Graz, Austria; nariae.baik@medunigraz.at (N.B.-S.); bernhard.schwaberger@medunigraz.at (B.S.); lukas.mileder@medunigraz.at (L.M.); nina.hoeller@medunigraz.at (N.H.); berndt.urlesberger@medunigraz.at (B.U.); 2Research Unit for Neonatal Micro- and Macrocirculation, Division of Neonatology, Medical University of Graz, 8036 Graz, Austria; 3Research Unit for Cerebral Development and Oximetry, Division of Neonatology, Medical University of Graz, 8036 Graz, Austria; 4Institute for Medical Informatics, Statistics and Documentation, Medical University of Graz, 8036 Graz, Austria; alexander.avian@medunigraz.at

**Keywords:** cardiac output, cerebral oxygenation, term neonates, neonatal transition

## Abstract

The immediate transition from foetus to neonate includes substantial changes, especially concerning the cardiovascular system. Furthermore, the brain is one of the most vulnerable organs to hypoxia during this period. According to current guidelines for postnatal stabilization, the recommended parameters for monitoring are heart rate (HR) and arterial oxygen saturation (SpO_2_). Recently, there is a growing interest in advanced monitoring of the cardio-circulatory system and the brain to get further objective information about the neonate’s condition during the immediate postnatal transition after birth. The aim of the present study was to combine cardiac output (CO) and brain oxygenation monitoring in term neonates after caesarean section in order to analyse the potential influence of CO on cerebral oxygenation during neonatal transition. This was a monocentric, prospective, observational study. For non-invasive cardiac output measurements, the electrical velocimetry (EV) method (Aesculon Monitor, Osypka Medical, CA, USA) was used. The pulse oximeter probe for SpO_2_ and HR measurements was placed on the right hand or wrist. The cerebral tissue oxygen index (cTOI) was measured using a NIRO-200NX monitor with the near-infrared spectroscopy (NIRS) transducer on the right frontoparietal head. Monitoring started at minute 1 and was continued until minute 15 after birth. At minutes 5, 10, and 15 after birth, mean CO was calculated from six 10 s periods (with beat-to-beat analysis). During the study period, 99 term neonates were enrolled. Data from neonates with uncomplicated transitions were analysed. CO showed a tendency to decrease until minute 10. During the complete observational period, there was no significant correlation between CO and cTOI. The present study was the first to investigate a possible correlation between CO and cerebral oxygenation in term infants during the immediate neonatal transition. In term infants with uncomplicated neonatal transition after caesarean section, CO did not correlate with cerebral oxygenation.

## 1. Introduction

To standardize the assessment of the condition of newborns during the immediate transition period after birth, Virginia Apgar developed a scoring system [1] that is nowadays widely used all over the world. However, there is significant inter- and intra-observer variability when the Apgar score is used [2,3]. In order to improve postnatal assessment, the latest guidelines recommend, besides clinical evaluation, monitoring of heart rate (HR) and arterial oxygen saturation (SpO_2_) with pulse oximetry and optionally with electrocardiography (ECG) in the delivery room [4]. However, these monitoring methods do not provide comprehensive information about potentially compromised cardio-circulatory status resulting in compromised oxygen delivery to various organs. One of the most vital organs certainly is the brain [5]. Oxygen delivery to the brain depends on the oxygen content of the blood (haemoglobin concentration and oxygen saturation) and cerebral perfusion. Cerebral perfusion depends on cardiac output (CO) and vascular resistance, whereby the evaluation of these cardio-circulatory parameters in the first minutes after birth remains challenging. Concerning cardio-circulation, centiles of HR during the first minutes after birth have already been published [6]. Our study group recently published reference ranges for blood pressure in the first minutes after birth [7]. Furthermore, we proved the feasibility of non-invasive CO measurements in term neonates during neonatal transition in the delivery room using the electrical velocimetry (EV) method [8]. The aim of the present study was to investigate whether there is a significant correlation between non-invasively monitored CO and cerebral oxygenation in term neonates during the neonatal transition period. We hypothesized that higher CO might be correlated with higher cerebral oxygenation.

## 2. Materials and Methods

This was a monocentric, prospective, observational study conducted from September 2013 to March 2017 at the Division of Neonatology, Department of Paediatrics and Adolescent Medicine, Medical University of Graz. The study was approved by the Regional Committee on Biomedical Research Ethics (EC number: 25-342 ex 12/13). Informed parental consent was obtained antenatally before neonates were included in the study.

Neonates born by caesarean section were included. After the neonates were fully delivered, a stopwatch was started. After cord clamping, which was routinely performed within 30 s, neonates were brought to the resuscitation table and placed under an overhead heater in the supine position. Measurements were performed during the first 15 min after birth. For non-invasive CO measurements, the Aesculon monitor (Osypka Medical, La Jolla, CA, USA) was used. Before starting the measurement, the skin was cleaned from vernix and the four surface electrodes were placed on the left forehead, left side of the neck, left hemithorax, and left thigh.

CO was calculated as an average out of six 10 s periods (with beat-to-beat analysis). The data from these 10 s periods were only accepted if the signal quality index (SQI) was ≥80%. The pulse oximeter probe for SpO_2_ and HR measurements (IntelliVue MP 30 Monitor, Philips, Amsterdam, The Netherlands) was placed on the right hand or wrist. The cerebral tissue oxygenation index (cTOI) was measured using a NIRO-200NX monitor (Hamamatsu Photonics, Hamamatsu, Japan). The near-infrared spectroscopy (NIRS) sensor was positioned on the right frontoparietal head and secured with a cohesive conforming bandage (Peha-haft, Harmann, Heidenheim, Germany).

Resuscitation was performed according to latest guideline recommendations [4,9]. We only analysed data from neonates who had uncomplicated neonatal transition periods (without any need for respiratory and/or medical support). All variables were stored using the multichannel system alpha-trace digital MM (BEST Medical Systems, Vienna, Austria) for subsequent analyses. Values of SpO_2_ and HR were stored every second, and the sampling rate of cTOI was 2 Hz.

Baseline characteristics are presented as mean ± standard deviation (SD) for normally distributed continuous variables and medians with interquartile range (IQR) when the distribution was skewed. Categorical variables are given with numbers and percentages. Changes in CO (mL/kg/min), cTOI (%), SpO_2_ (%) and HR (beats per minute) were analysed using a linear mixed-effects model with a fixed effect for time and a first-order autoregressive covariance structure. Estimated mean scores with 95% confidence intervals (CI) are given for the analysed variables for 5, 10, and 15 min. To analyse possible associations between CO and cerebral oxygenation, correlation analyses (Spearman’s rank correlation coefficient) were performed separately at minutes 5, 10, and 15 after birth. A *p*-value < 0.05 was considered statistically significant. Statistical analyses were performed using IBM SPSS Statistics 26.0.0 (IBM Corporation, Armonk, NY, USA).

## 3. Results

During the study period, 99 term neonates were enrolled. The demographic and clinical characteristics of the study population are presented in Table 1.

### 3.1. Course of CO, cTOI, SpO_2_, and HR at Minutes 5, 10, and 15 after Birth

The courses of vital parameters at the time points 5, 10, and 15 min after birth are presented in Table 2. CO decreased from minute 5 to 10 and significantly increased afterwards. cTOI and SpO_2_ showed a statistically significant increase from minute 5 to 10 after birth. Additionally, SpO_2_ rose significantly from minute 10 to minute 15, but after considering the number of values, this increase is not relevant for clinical praxis. HR did not change from minute 5 to 10 or 15.

### 3.2. Correlation between CO and cTOI at minutes 5,10 and 15 after birth

There was no correlation between CO and cTOI during the first 15 min after birth (Table 3/Figure 1 and Figure 2).

## 4. Discussion

To our knowledge, this is the first study presenting correlation analyses between CO and cerebral oxygenation in term infants during the immediate neonatal transition. As the brain is a vulnerable organ, it is important to better understand oxygen delivery to the brain during this time. We hypothesized that higher CO would result in higher cerebral oxygenation, as oxygen delivery is dependent on CO and oxygen content. However, in our study population of 99 term infants delivered by caesarean section, we could not observe any significant correlation between CO and cerebral oxygenation.

Most published haemodynamic changes during the immediate neonatal transition described an increase in heart rate and a decrease in pulmonary vascular resistance after cord clamping and lung aeration [10,11,12,13,14,15]. Some research groups have published data showing changes in cardiac output in recent studies, mostly using echocardiography [13,14]. In the present study, we observed a significant decrease in cardiac output from minute 5 (199 mL/kg/min) until minute 10 (187 mL/kg/min). Afterwards, CO rose again until minute 15 (198 mL/kg/min) after birth. In a study using echocardiography, Van Vonderen et al. [15] described an increase in cardiac output from minute 2 (151 mL/kg/min) to minute 5 (203 mL/kg/min), and afterwards a stable tendency until minute 10 (201 mL/kg/min). A further study described an increasing tendency of cardiac output in the first 20 min after birth, which did not reach significance. CO was 168 mL/kg/min at p1, 186 mL/kg/min at p2, and 189 mL/kg/ min at p3 [10]. These findings are comparable with the results in the present study.

Compared with echocardiography, EV provides the opportunity to monitor CO continuously and objectively. Additionally, the feasibility of this method has already been proven in the delivery room in term infants after caesarean section [8], and in vaginally born neonates receiving delayed cord clamping [9].

Noori et al. [10] compared EV measurements with echocardiography in 20 healthy neonates during the first two days after birth. They performed left ventricular output measurements with EV and echocardiography. There was no significant difference between these two methods, confirming that EV measurements have comparable accuracy and precision to echocardiography for CO measurements when the EV SQI was ≥80% [10].

Several studies have described lower cerebral oxygenation values in preterm neonates immediately after birth, resulting in cerebral injury in the following days [11,12]. As a recent study described a possible association between lower left ventricular output measured by echocardiography and the development of intraventricular haemorrhage during the first day of life in extremely preterm infants [13], the authors raised the question of whether CO immediately after birth has an impact on cerebral oxygenation. As a first step, we wanted to investigate whether there is an impact of CO on cerebral oxygenation in term infants during neonatal transition. However, in contrast to our hypothesis, we did not find such a correlation in healthy term neonates. Nonetheless, this finding is in accordance with a recent observation in term neonates that mean arterial blood pressure had no significant impact on cerebral oxygenation [16]. In contrast, there was a significant association between blood pressure and cerebral oxygenation in preterm neonates during neonatal transition [16]. These contradictory results may be a sign of intact vascular autoregulation in term neonates and potentially impaired autoregulation in preterm neonates. In the present study of term infants during uncomplicated transition without any need for respiratory and/or medical support, our results further emphasize the intact cerebral autoregulation in neonates born at term. With this knowledge, the next step should be CO monitoring in compromised term and preterm neonates to better understand the underlying (patho)physiological processes in those infants.

We recognize several limitations in this study. First, we only included neonates born by caesarean section. Due to technical reasons, not all of the measurements were able to be performed in the delivery room next to the mother. Since all neonates after caesarean section were brought to the resuscitation table and observed by a neonatologist for 10 to 15 min, we only performed measurements on those neonates. We did not include vaginally born neonates to avoid delaying immediate bonding with the mother. Therefore, we do not have any information about cardiac output measurements in vaginally born neonates. Recent studies showed that the mode of delivery might influence haemodynamic parameters and cerebral oxygenation during neonatal transition [6,17,18]. We may have observed some differences in vaginally born neonates. Additionally, in these neonates, cord clamping time was within 30 s, as this was the routine procedure during the observational period. Delayed cord clamping might have resulted in better oxygenation in these neonates. Second, our study population contains exclusively term neonates. In our study population, we observed stable values concerning cardiac output during the immediate neonatal transition. The next interesting question would be whether there are observable differences in very low birth weight preterm neonates needing extended respiratory and/or medical support. Third, since we only accepted and analysed cardiac output measurements, if the signal quality index was >80%, to guarantee the reproducibility with the other already approved methods, we had to exclude 76% of NICOM measurements. In many cases, the activity and movements of the neonates influenced the NICOM measurements. Further studies are needed before introducing this method into routine clinical practice.

## 5. Conclusions

The present work is the first study to investigate a possible correlation between CO and cerebral oxygenation in term infants during immediate neonatal transition. In term infants with uncomplicated neonatal transition after caesarean section, there was no significant correlation between CO and cerebral oxygenation at 5, 10, or 15 min after birth.

## Figures and Tables

**Figure 1 children-08-00439-f001:**
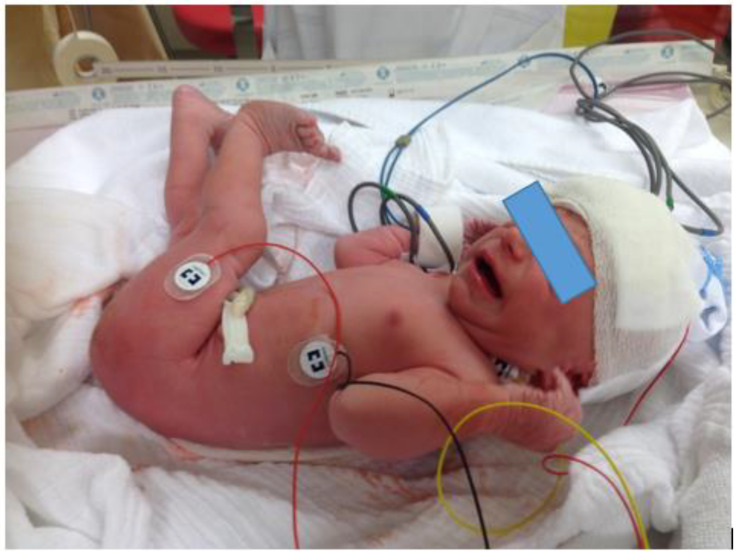
Neonate with four surface electrodes on the left forehead, left side of the neck, left hemithorax, and left thigh for non-invasive CO measurement.

**Figure 2 children-08-00439-f002:**
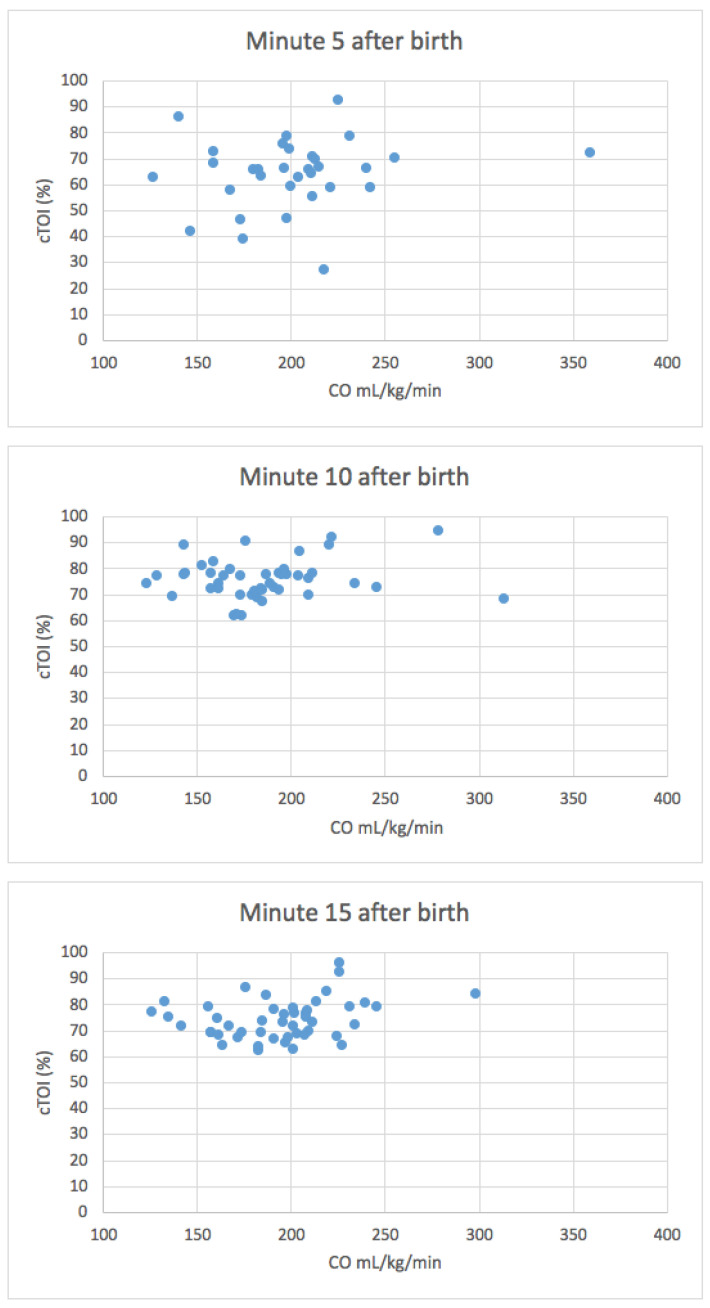
Correlation analyses between CO and cTOI at minute 5, 10, and 15 after birth.

**Table 1 children-08-00439-t001:** Demographic and clinical characteristics of the study population.

	Study Population (*n* = 99)
Gestational age (weeks)—median (IQR)	38.8 (38.3–39.3)
Birth weight (g)—mean (SD)	3296 ± 492
Body length (cm)—median (IQR)	51 (49–52)
Head circumference (cm)—median (IQR)	35 (34–36)
Female sex—*n* (%)	54 (54.5)
Apgar score at 5 min—median (IQR)	10 (10–10)
Apgar score at 10 min—median (IQR)	10 (10–10)
Umbilical artery pH—median (IQR)	7.29 (7.27–7.31)
Maternal spinal anaesthesia—*n* (%)	99 (100%)

**Table 2 children-08-00439-t002:** Courses of vital parameters at 5, 10, and 15 min after birth.

	Minute 5 after Birth	Minute 10 after Birth	Minute 15 after Birth	*p*-Value Minute 5 to 10/Minute 10 to 15
cTOI (%) mean (95% CI)	63.8 (61.9–65.8)	73.2 (71.3–75.0)	73.1 (71.2–75.0)	<0.001/0.935
SpO_2_ (%) mean (95% CI)	81.2 (79.6–82.8)	93.4 (91.9–95.0)	95.2 (93.6–96.8)	<0.001/0.024
HR (beats per minute) Mean (95% CI)	152 (148–156)	151 (147–155)	153 (149–157)	0.618/0.308
CO (mL/kg/min) Mean (95% CI)	199.8 (188.3–211.4)	187.5 (177.3–197.7)	198.0 (187.7–208.3)	0.019/0.019

**Table 3 children-08-00439-t003:** Correlation analyses between CO and cTOI.

		cTOI
**Cardiac output**	Minute 5 after birth	
	*ρ*	0.170
	*p*-value	0.351
	Minute 10 after birth	
	*ρ*	0.074
	*p*-value	0.629
	Minute 15 after birth	
	*ρ*	0.265
	*p*-value	0.071

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
