# Peer review of "Cardiac Output and Cerebral Oxygenation in Term Neonates during Neonatal Transition"

_children, 2021, doi:10.3390/children8060439_

Round 1
Reviewer 1 Report
This is a well-written research article evaluating the correlation between non-invasively monitored cardiac output (CO) and cerebral oxygenation. The concept is novel and the study design is appropriate.
Specific comments:
1.Introduction:
Line 59-61- The aim and hypothesis of the study may be reworded to better depict what the authors have evaluated. The authors demonstrate the correlation between CO measurements and cerebral oxygenation but cannot prove cause and effect in this model. The authors describe the size and direction of a relationship between the variables but cannot suggest that change in one causes (or "results in") the change in the other.
2.Materials and Methods:
Line 69-71- Why was deferred cord clamping not performed for at least 30-60 seconds? What is the current local recommendation? Benefits of deferred cord clamping (DCC) in term and preterm infants not requiring resuscitation has been proved and additionally DCC allows better oxygenation in the newborn (Dawson curves with higher SpO2 ranges with DCC). For future applicability of this study’s findings, it may be more appropriate to have performed DCC. This could be added as a limitation.
Line 77- The image caption should be moved to the previous page. Also, this could be labeled Figure 1 and the subsequent figure in the results could be labeled as Figure 2.
Line 81- Why was signal quality index >80% chosen, rather than >90% or 95%? Is there a reference?
3.Results:
Line 112-113- Since the observation period begins from 4-5 min (6 observations of 10 sec each), cTOI and SpO2 showed statistically significant increase during whole observation periods at 5, 10 and 15 min after birth, not the whole 15 min after birth.
Line 116- Again, the reported results are for 5, 10, and 15 min time-points averaged over a min each, but not the whole 15 min after birth.
Line 164- “term” should be removed from this sentence.
Line 169- For the next step, could CO and blood pressure both be monitored in compromised term and preterm infants, along with cerebral oxygenation? Would that give additional valuable information?
Line 178-180- Were these cesarean sections performed under spinal or general anesthesia? Maternal general anesthesia may affect neonate’s hemodynamic measurements as well.
Line 180-181- What is the basis behind the statement “we might have observed some difference in vaginally born neonates”?
Line 182- 185- The sentence structure is confusing- this sentence may be reworded.
Line 185- 188- What percentage of NO-COM measurements were excluded?
Reviewer 2 Report
Study by Baik-Schneditz et al is novel, as it involves non-invasively measuring cardiac output in neonates soon after birth. The authors did not a correlation between cardiac output and cTOI in term neonates. The reviewer has the following comments regarding the study.
Page 6, L134-136. The data on CO at 5 minutes and 10 minutes is not correlating with the data presenting on Table 2 data. (194 / 191 versus 199.8 / 187 in Table 2). Also, the authors have commented on no significant difference in CO between 5 and 10 minutes in the discussion. Table 2 demonstrates significant difference between 5 and 10 minutes CO data.
Would the authors explain in the discussion, why the CO decreased from 5 to 10 minutes and again why it increased at 15 minutes. Is there any literature on trends in CO soon after birth?
The CO at 5 minutes and 15 minutes is essentially the same. However, cTOI increased from 5 to 15 minutes. How would the authors explain the increase in cTOI in the presence of no corresponding increase in CO. That discussion is needed.
SpO2 and cTOI both increased at 10 minutes compared to 5 minutes, with a further increase in SPO2 at 10 minutes. What would account for an increase in SPO2. Also discuss the factors that lead to an increase and SPO2 and cTOI son after birth.
The authors should include a discussion on the limited accuracy of EIV (electrical impedance velocimetry) in the measurement of CO (Feng SM & Jin Liu, 2020; Mekis SD, Kamenik M et al, 2008) compared to echocardiography or thermodilution methods.
